# Stroke but no hospital admission: Lost opportunity for whom?

**Carine Milcent**[1]*, **Hanta Ramaroson**[2], **Fleur Maury**[3], **Florence Binder-Foucard**[4], **Marie Moitry**[5], **Anne-Marie Moulin**[6]

1 Center for National Scientific Research – CNRS, Paris School of Economics – PSE, Health Economics, Econometrics, Paris, France, 2 Medical Information Analysis and Coordination Unit (UCAIM)- Medical Information Department, Bordeaux University Hospital Centre, Bordeaux, France, 3 Medical Information Department, Lille University Hospital, Lille, France, 4 Public Health Department, Strasbourg University Hospital, Strasbourg, France, 5 Department of Epidemiology and Public Health, University of Strasbourg, Strasbourg, France, 6 CNRS, SPHERE (CNRS-Université de La Sorbonne-Paris), Paris, Cedex 13, France

* carine.milcent@psemail.eu

**Data Availability Statement:** The data underlying the results presented in the study is third party data. Restrictions apply to the availability of these data, which were used under license for this study. Data used for this study are reported to the

## Abstract

To counter the spread of COVID-19, the French government imposed several stringent social and political measures across its entire population. We hereto assess the impact of these political decisions on healthcare access in 2020, focusing on patients who suffered from an ischemic stroke. We divide our analysis into four distinct periods: the pre-COVID-19 pandemic period, the lockdown period, the "in-between" or transitional period, and the shutdown period. Our methodology involves utilizing a retrospective dataset spanning 2019–2020, an exhaustive French national hospital discharge diagnosis database for stroke inpatients, integrated with income information from the reference year of 2019. The results reveal that the most affluent were more likely to forgo medical care, particularly in heavily affected areas. Moreover, the most disadvantaged exhibited even greater reluctance to seek care, especially in the most severely impacted regions. The data suggest a loss of opportunity for less severely affected patients to benefit from healthcares during this lockdown period, regardless of demographic, location, and socioeconomic determinants. Furthermore, our analysis reveals a notable discrepancy in healthcare-seeking behavior, with less affluent patients and seniors (over 75 years old) experiencing slower rates of return to healthcare access compared to pre-pandemic levels. This highlights a persistent gap in healthcare accessibility, particularly among socioeconomically disadvantaged groups, despite the easing of COVID-19 restrictions.

## 1. Introduction

The COVID-19 pandemic has triggered an unprecedented worldwide response aimed at curbing the spread of the virus. This response has resulted n the implementation of political measures designed to mitigate its impact on the general population's health and to preserve healthcare systems that have come under significant strain. In particular, hospital departments underwent major reorganizations during the initial wave of the pandemic As a consequence,

National Data Protection Authority. The data supporting this study's findings are from Agence Technique d'Information Hospitalière (ATIH) Digital. The website to use when requesting access for the database is (https://www.health-data-hub. fr). The authors confirm that others can access these datasets and confirm that others would be able to access these data in the same manner as the authors. The study was exempt from informed consent requirements because the dataset contained no information to identify patients.

**Funding:** Research grant from the Research French Agency ANR - PEPR Digital Health under the number ANR-22-PESN-0005.

**Competing interests:** NO authors have competing interests.

marked reductions in hospital admissions have been described for many urgent or elective non-COVID activities [1–6], including hospitalizations for stroke and stroke-related revascularization procedures [7–17]. Such declines may be attributed to patients' fear of exposure to COVID-19, the government's directives to stay at home, or the effects of social isolation. Patients with neurovascular symptoms may have delayed or avoided seeking medical attention [18,19], potentially explaining the drop in stroke alert activations reported in many countries [12,20]. For ischemic events in particular, this could have significant ramifications in terms of missed opportunities, as prompt pre- and in-hospital management are crucial to improve patients' outcomes [21–23].

Social and educational inequalities impact health disparities [24]. Several studies have established a correlation between education, socioeconomic status, and cardiovascular disease [19,20]. Concerning stroke, social deprivation accompanies a poorer understanding of stroke risks and symptoms [21], as well as a higher incidence and severity during the acute phase, along with lower quality of care and prognosis [22,23,25–27]. As shutdowns and lockdowns appear to have a disproportionate impact on the most disadvantaged populations [28,29], disparities in health outcomes and in healthcare seeking behavior according to socioeconomic status may have either diminished or intensified in France during this period, and, by extension, in the entirety of 2020. This exceptional year was marked by fluctuations in viral circulation rates, with early signals of the first wave between weeks 7 and 9. The municipal elections delayed the decision to implement the national lockdown decision until week 11. Warnings of a second wave emerged around week 41, prompting the authorities to use the national school vacations to enforce a national shutdown in the fall.

In this paper, employing a highly innovative method, we explore the variations in data for inpatient admittance for stroke between 2019 and 2020 in France. We then assess how the factors of geographical location, vulnerability, and deprivation influence healthcare-seeking behavior in function of the intensity of the COVID-19 pandemic and the policy decisions implemented to contain and control the spread of the virus.

This paper follows a standard formula. Section 2 presents the database and the methodology. Section 3 outlines the empirical framework. Section 4 exhibits the results, which are discussed in Section 5. Finally, Section 6 concludes the paper.

## 2. Data

We conducted a population-based retrospective study of patients admitted to acute care units for ischemic stroke stays across all French hospitals during the period of 2019–2020. The selection criteria for the population were based on the International Classification of Diseases, 10th revision (ICD-10), with the corresponding codes being I630-I636, I638, and I64, as established in previous literature [30–32]. To follow, we detail the data sources utilized in this study.

### 2.1 Data availability statement

The data underlying the results presented in the study is third party data. Restrictions apply to the availability of these data, which were used under license for this study. Data used for this study are reported to the National Data Protection Authority. The data supporting this study's findings are from Agence Technique d'Information Hospitalière (ATIH) Digital. The website to use when requesting access for the database is (https://www.health-data-hub.fr).

The authors confirm that others can access these datasets and confirm that others would be able to access these data in the same manner as the authors. The study was exempt from informed consent requirements because the dataset contained no information to identify patients.

## 2.2 Data collection

Data were extracted from three different sources:

- French National Acute Care Hospital Discharge Diagnosis Databases _ PMSI-MCO for 2018, 2019, and 2020. They contain information regarding the patient's demographic characteristics (age, sex) as well as the primary diagnosis, comorbidity factors, and complications based on ICD-10. Databases contain information regarding the patient's social vulnerability as ICD-10 coded. Databases also include the postcode of the individual's residence.

- French Fiscal and Income Data _ INSEE-IRCOM database from the French National Statistics. This database is open-access at the postcode level. Information on unemployment, level of education, and fiscal details, which are integrated into the selected stays within the PMSI-MCO database, using the postcode as the identifier.

- French Deprivation Index Database contained within the INSEE-INSERM database. We utilize the French Deprivation Index (FDep), which provides information at the sub-municipal level. The European Deprivation Index inspires this variable, which has been developed by [33].

## 2.3 Computed variables

From these three databases, we computed the following variables.

- From the PMSI-MCO database: Age group categorized by age range ($<$ 65 years; 65 to 74 years; 75 to 84 years; $>$ 84 years).

- Complications, categorized into three types according to either primary or related diagnostic codes registered during the stay:

  - inhalation pneumonia (ICD-10: J690, J698)

  - thrombosis (ICD-10: I801, I802, I803, I808, I809)

  - pulmonary embolism (ICD-10 I260, I269).

- Consequences, categorized into two types according to diagnostic codes (main or related) registered during the stay:

  - aphasia (ICD-10: R47.00, R47.01, R47.02)

  - paralysis (ICD-10: G81.00, G81.01, G81.08, G81.1, G81.9, G82.0, G82.2, G82.3, G82.4, G82.5, G83.0, G83.1, G83.4, G83.8+0).

We also compute a dummy for the presence of chronic disease(s) as collected in the database.

In addition, from the patient's postcode, we establish a dummy variable for residences in rural areas. It is important to note that residing in a rural area does not inherently imply that the area is impoverished. However, regardless of the area's economic level, rural residents often face challenges accessing healthcare.

A dummy variable is created for enrollment in universal health protection, known as PUMa (Protection Universelle Maladie). This insurance is part of public health insurance system and is designed for people outside the labor market who are no longer seeking employment. It is important to acknowledge that not all eligible individuals avail themselves of the PUMa programs, for various reasons, such as difficulty asserting their entitlement to the

program. Additionally, individuals who are identified as vulnerable by healthcare teams may still be ineligible for this program. However, it is worth noting that the program aims to provide universal access to healthcare.

The extreme vulnerability dummy variable is defined as having at least one of these individual characteristics:

- Childhood severity issue (ICD-10 Z61),

- Social, environmental issue (ICD-10 Z60),

- Employment severity issue (ICD-10 Z56),

- Housing severity issue (ICD10-Z59),

- Educational severity issue (ICD-10 Z55).

The hospital staff codes this information using CMD-10 during the hospital stay only when the patient is in a highly critical situation. There is no financial incentive to code this information because no additional payment results from it. Therefore, the presence of this information indicates that the patient's economic or social context affects their medical needs. Table 1, last column, presents the percentage of the stroke population in extremely vulnerable situations, which was 7.5% in 2019. During the COVID-19 pandemic, we anticipated a decrease in

**Table 1. Evolution of the variation in healthcare-seeking behavior from 2019 to 2020.**

| Variation from 2019 to 2020 | | P1 | P2 | P3 | P4 | 2019 (%) |
|---|---|---|---|---|---|---|
| Age | <65 years | -1.15% | **-12.01%** | +3.53% | +1.89% | 24.0% |
| | 65 to 74 years | +3.03% | **-8.44%** | +2.81% | +3.36% | 22.0% |
| | 75 to 84 years | +1.00% | **-9.33%** | -2.66% | -3.35% | 26.9% |
| | >84 years | -3.37% | **-12.71%** | -3.19% | **-9.08%** | 26.1% |
| Health Status | No complication | -0.76% | **-11.48%** | -0.61% | -1.90% | 91.9% |
| | With Complications | -1.77% | **-1.43%** | 2.91% | -2.61% | 8.1% |
| | No consequence | 1.22% | **-14.25%** | 0.76% | -3.07% | 49.9% |
| | With consequences | -1.86% | **-8.15%** | -1.16% | -1.12% | 51.1% |
| | Chronic disease | +0.94% | **-12.85%** | -0.42% | **-4.76%** | 55.8% |
| Geographical residence | Suburb | +1.16% | **-10.71%** | +2.27% | **-4.70%** | 35.8% |
| | Downtown area | +1.20% | **-11.84%** | **-4.92%** | -2.00% | 22.1% |
| | Isolated city | -2.14% | **-15.13%** | +0.82% | +0.38% | 5.0% |
| | Rural area | -0.85% | **-9.94%** | +0.44% | -1.79% | 37.0% |
| Vulnerability index | Extreme vulnerability | **+4.23%** | **+5.16%** | +2.98% | +0.55% | 7.5% |
| | No extreme vulnerability | -0.41% | **-11.84%** | **+5.96%** | **-2.44%** | 92.5% |
| Deprivation index | Above Q1 | **-3.43%** | **-12.04%** | **+5.70%** | **+4.92%** | 19.4% |
| | Q1-Q2 | -0.91% | **-10.78%** | +0.05% | +0.96% | 21.86% |
| | Q2-Q3 | -0.08% | **-8.94%** | -1.43% | -2.97% | 22.09% |
| | Q3-Q4 | +1.46% | **-9.18%** | -2.70% | **-3.88%** | 20.24% |
| | Over Q4 | +1.88% | **-15.68%** | -3.77% | -5.91% | 16.43% |

Source: PMSI-MCO, INSEE-IRCOM, INSEE-INSERM, exhaustive administrative database for stroke patients _ 2019–2020.

Note: P _ period, from 1 to 4.

P1: 01/01/2020 to 16/03/2020.

P2: 17/03/2020 to 09/05/2020.

P3: 10/05/2020 to 15/11/2020.

P4: 16/11/2020 to 17/12/2020.

behavior coding for these codes due to a reduction in available time. however, we will see later in this paper that this was not observed. Here, we identify individuals in the most dire situations, including individuals who usually escape public statistics because they are homeless.

From the INSEE-IRCOM database, we use information aggregated at the postcode level, including median income and rates of single parenting, single-occupancy households, level of employment, lack of secondary education, lack of tertiary education, and unemployment.

From the INSEE-INSERM database, deprivation indicators (Fdep) were defined in 2018 at a granular geographical level for the entire French population. This index is constructed according to information from public statistics, excluding the most severely impoverished part of the population, such as those experiencing homelessness. We sum up the deprivation indicators (Fdep) to compute the index at the postcode level. We then compute the distribution quintiles of this information.

### 2.4 Specific computed variables for the COVID-19 pandemic

In France, the COVID-19 pandemic in 2020 can be divided into four periods ($P_1$ to $P_4$), according to intensity. As in the paper on mental health effects of the COVID-19 pandemic as well as elsewhere [34], we consider two major waves over the year.

The first period ($P_1$) is the pre-COVID-19 pandemic period when the government and the health authorities turned their attention to the daily number of deceased people due to the COVID-19 virus, and the population was still hesitant about the health impact of this COVID-19 virus.

The second period ($P_2$) corresponds to the lockdown phase intended to curb the spread of COVID-19 and to safeguard the French healthcare system. This period started on 17 March and continued until 10 May 2020. A stringent nationwide lockdown marked this period. Mobility was severely restricted, with individuals instructed to stay home and work remotely. Police checkpoints were established, and fines were strictly imposed. The lockdown applied to all offices, shops, colleges, schools, and institutions considered non-essential.

The third period ($P_3$), the "in-between" (transitional) period, marks the end of the lockdown when social gatherings were discouraged, and social distancing was still the norm.

The fourth period ($P_4$) is the shutdown period from mid-October, with the start of the Fall school vacation. Working remotely was highly advised but no longer imposed. Schools stayed open, and a broader definition of essential shops was applied. This period ended with the New Year's holiday season.

This study covers the years 2019 and 2020, with 2019 as the reference year.

During the lockdown, the progression of COVID-19 across the country was heterogeneous, resulting in varying degrees of pressure on the healthcare system. The east of France and the Paris region experienced critical situations with intensive care units overwhelmed by COVID-19 patients, whereas the regions in the north were only minimally impacted. The regions in the west and southwest were initially spared. However, during the second period of the COVID-19 spread, some regions previously bypassed by the pandemic were, in turn, affected.

To illustrate this regional disparity, we created a binary variable for each period (*regional crisis*) equal to one when the patient's residence was in a critical situation area. The detailed information is given Table A6 in S1 Appendix.

## 3. Empirical framework

The following data were prospectively collected and retrospectively analyzed.

This paper assumes that, in the absence of the COVID-19 pandemic, the distribution of stroke patients in 2020 would have followed a pattern similar to that in 2019. As a

comparative analysis, we checked that this is observed when comparing 2018 to 2019. The methodology described below was applied in the years 2018–2019. The differences of in-admission rates from 2018 to 2019 are negligible. These differences are used as a benchmark to judge the 2019–2020 differences. Results for the years 2018 and 2019 are presented in the S1 Appendix.

### 3.1 A non-parametric method

We employed a non-parametric method to assess the percentage of non-admitted individuals with stroke who would have been admitted before the COVID-19 pandemic restriction period, taking into account the patient's clinical characteristics as well as his or her socioeconomic characteristics. The construct is the following:

Patients were grouped (denoted as group $i$) according to a set of variables.

For each group $i$ defined, four periods corresponding to the phases of the COVID-19 pandemic were considered. These subgroups$_{ij}$ are computed for the year 2019 (2019-subgroup$_{ij}$) and for the year 2020 (2020- subgroup$_{ij}$).

Each subgroup$_{ij}$ is defined by the period $P_j$ crossed with the year. In the regression results presented here, the subgroups of patients are computed according to the following dummy variables: over 74 years old, with complications, with consequences, with vulnerability, rural areas, chronic diseases, quintile of the French deprivation index (Fdep) for the most deprived (Q5), for the deprived (Q4), for the better-off (Q1), and region with local COVID-19 pandemic.

There are $2^8 = 256$ groups$_i$, subdivided into four according to the period $P_j$. Therefore, there are 1'024 subgroups$_{ij}$.

Sensitivity analyses, were conducted running models with alternative sets of variables to set up groups of patients. Results are available upon request.

Next, for each subgroup$_{ij}$, we computed the variation (%) in the number of inpatients between the two years (2019 and 2020), with 2019 serving as a reference. Using variation allows us to eliminate the magnitude of the difference in the number of inpatients across groups. We introduce weightings in the model to distinguish strong variations exclusively for marginal groups of patients in the distribution. This variation$_{ij}$ is then weighted by the size of the 2019-subgroup$_{ij}$ in the distribution. As a sensitivity analysis, we selected different sets of variables for group compositions. According to Milcent [35], the location is one of the variables affecting hospital access.

A limitation for this method is the inability to calculate standard deviations. Indeed, for each subgroup$_{ij}$, we compute a unique sum. Therefore, the only way to evaluate the validity of the results is by comparing these figures with figures obtained for the years 2018 to 2019. Furthermore, we lack statistical tests to assess the impact of deprivation and vulnerability factors on healthcare access behavior. To address these challenges, we use a parametric weighted least squares model.

### 3.2 A weighted ordinary least square

As a second step, we study the characteristics of the groups of individuals with non-admitted stroke who would have been admitted before the COVID-19 pandemic. The method is based not on individual-level information but on aggregated data to obtain information on the number of individuals at risk in each subgroup$_{ij}$. We use weighted least squares to estimate differences between years and periods. The subgroup size, i.e., individuals at risk of having a stroke

during a given period in 2019, is defined as the weights.

$$Y_{ij} = \alpha + \beta_1 \% d(over\ 85)_{ij} + \beta_2 \%\ d(with\ complication)_{ij} + \beta_3 d(rural\ area)_{ij}$$
$$+ \beta_4 d(vulnerability\ index)_{ij} + \beta_5 d(Deprivation\ index)_{ij} + \varepsilon_{ij}$$

(1)

d(.): dummy variable defined for (.).

The dependent variable $Y_{ij}$ is the variation in healthcare access for the subgroup$_{ij}$. In other words, we computed the variation in the number of inpatients in subgroup$_{ij}$ over the two years ((subgroup$_{ij2019}$—subgroup$_{ij2020}$)/ subgroup$_{ij2019}$). The greater the value of $Y_{ij}$, the greater the reduction in healthcare access. This reduction can be due to self-renunciation, a postponed decision., or may even be due to the inability to be admitted to healthcare centers. However, the urgent nature of a stroke imposes a significant limitation on this interpretation. $\varepsilon_{ij}$ is a random error term.

Standard errors are clustered at the regional level to correct for within-region correlation in outcomes. However, coefficients were estimated using weighted ordinary least square, with region-fixed effect when specified: the region-fixed effect was included to control for time-invariant unobservable region characteristics that may affect outcomes [36,37].

Model A focuses on comparing the better-off areas to those with a poorer deprivation index. The dummy variable for the *Deprivation index$_{ij}$* is equal to 1 when the deprivation index is higher or equal to Q1, zero otherwise.

In Model B, the dummy variable for *Deprivation index$_{ij}$* is equal to 1 when the deprivation index is equal or lower to Q4, zero otherwise.

In Model C, the dummy variable for *Deprivation index$_{ij}$* is equal to 1 when the deprivation index is equal to or lower than Q5, zero otherwise.

Models D and E present incremental regressions. We add a variable to control for the percent of inpatient stroke events with chronic disease (Model E).

We also suggest examining the socioeconomic gradient effect within areas experiencing a regional crisis by incorporating crossed variables (regional crisis dummy crossed with socioeconomic gradient). As is standard in DiD models, identification relies on the "common trend assumption" that, in the absence of the policy, outcomes in the "treated" region would have evolved as in the "untreated" regions.

$$Y_{ij} = \alpha + \beta_1 \%\ d(over\ 85)_{ij} + \beta_2 \%\ d(with\ complication)_{ij} + \beta_3 d(rural\ area)_{ij} + \beta_4 d(vulnerability\ index)_{ij}$$
$$+ \beta_5 d(Deprivation\ index)_{ij} + \beta_5 \left[ d(Deprivation\ index)_{ij} * d(vulnerability\ index)_{ij} \right] + \varepsilon_{ij}$$

(2)

All analyses were performed with SAS Enterprise version 7.1 and Stata version 16. The research was conducted by applying a significance level of 5%. The data that support the findings of this study are available from *Agence Technique d'Information Hospitalière* (ATIH) Digital. Informed consent is not required since the study was based on routinely collected de-identified administrative data, as regulated by French law. Data used for this study are reported to the National Data Protection Authority. Restrictions apply to the availability of these data, which were used under license for this study.

## 4. Results

### 4.1 Statistical analysis

Our study sample consisted of 214,720 patients admitted for ischemic stroke in the years 2019 and 2020. Fig 1 depicts the weekly number of strokes during the four periods outlined in our analysis. This figure illustrates the variation in healthcare-seeking behavior for stroke

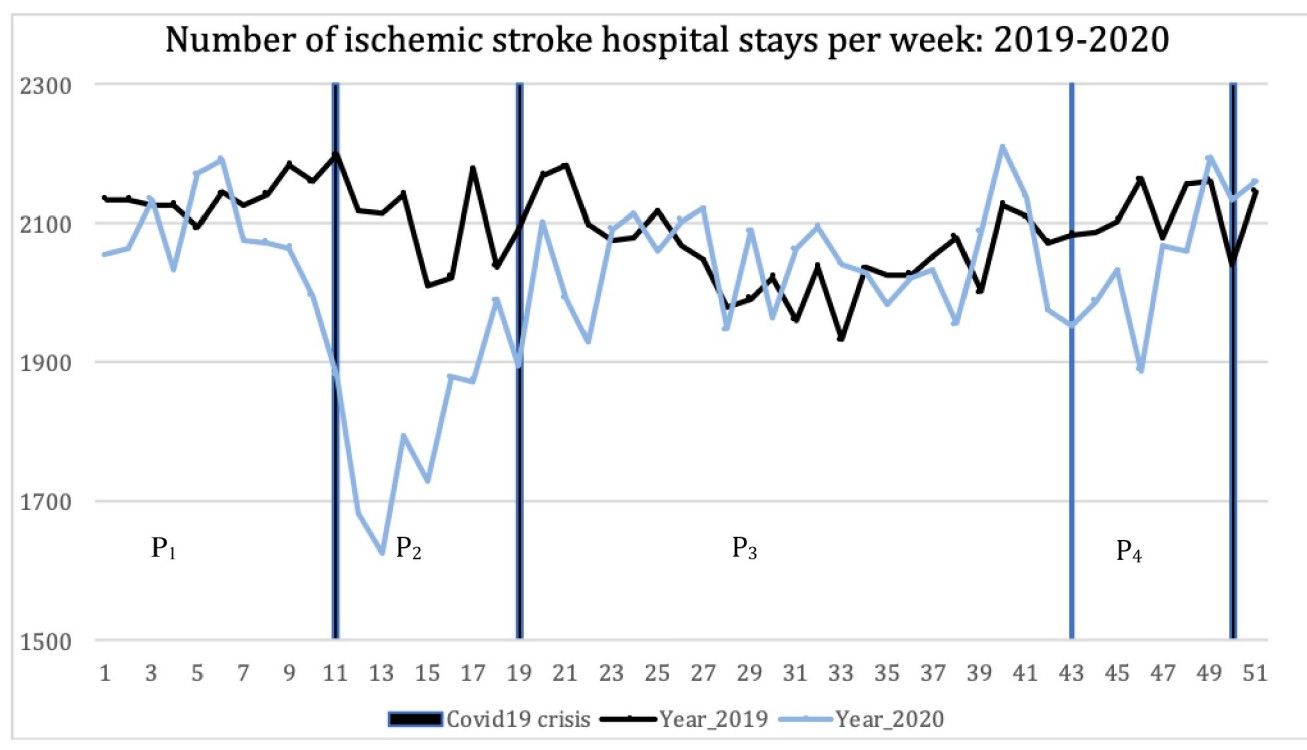

**Fig 1. Evolution of the number of inpatients for stroke per week _ 2019 versus 2020: Acute care hospitalization.**

| Period | Pre-COVID-19 pandemic: P₁ | | Lockdown: P₂ | "In between"P₃ | Shutdown: P₄ |
|---|---|---|---|---|---|
| | Before any rumors | With lockdown rumors (3 weeks before the lockdown) | | | |
| Variation in ischemic stroke | -0.85% | -4.65% | | | |
| | | -2,38% | -14.07% | -0.68% | -2,85% |

*Source: PMSI-MCO, exhaustive administrative database for stroke patients _ 2019-2020*
*Note: Stay at the date of admission*
P: period, from 1 to 4.
P1: 01/01/2020 to 16/03/2020
P2: 17/03/2020 to 09/05/2020
P3: 10/05/2020 to 15/11/2020
P4: 16/11/2020 to 17/12/2020

throughout 2020. There was a sharp decline in stroke admission during the lockdown period ($P_2$), beginning three weeks prior to the lockdown decision (the last three weeks of the $P_1$). During the "in-between" period ($P_3$), we observe a return to the 2019 distribution, with no rebound in stroke admission activity following the decline observed during the lockdown period. During the shutdown period ($P_4$), a slight decrease in inpatients' stroke admission is observed.

In 2019, approximately three-quarters of all strokes occurred in patients over 65. however, in 2020, we observed some changes in the stroke inpatients' case-mix distribution:

There were significant differences regarding stroke age as the share of patients over 85 years old: 27.1% in 2019 versus 26.1% in 2020 (T-test: $Pr(|T| > |t|) = 0$, P-value = 0.00), thrombosis

as complications with 1.43% in 2019 versus 1.58% in 2020 (T-test: $\Pr(|T| > |t|) = 0$, P-value = 0.01).

Changes were observed in the socio-economic conditions for stroke patients admitted: particularly in the high vulnerability index category, which increased from 5.42% in 2019 to 6.06% in 2020 (T-test: $\Pr(|T| > |t|) = 0$, P-value = 0.00). The differences in the distribution shares before and after the COVID-19 pandemic highlight the distribution variation over time. However, a change in the number of one group in the distribution can affect the other groups' share, even if there were no changes in number in the other groups. Computing the variation of distribution shares before and after the COVID-19 pandemic allows us to avoid this drawback.

## 4.2 Variation (%) in the number of inpatients over the two years

Table 1 shows the results of the variation in admission in 2020 compared to 2019 for groups of determinants. These determinants include demographic and geographic characteristics, clinical factors, deprivation and vulnerability drivers, and the variation of individuals not admitted in 2020 who had been admitted in 2019 in the absence of the COVID-19 pandemic. The values and their signs allow us to identify the reduction in healthcare access. Given that strokes require emergency hospital admission, even during the COVID-19 pandemic, this reduction in healthcare access can be interpreted as non-sought care behavior. The variation in healthcare-seeking behavior is presented for each of the four periods across various determinants. By setting this up, we do not have a standard deviation. Indeed, the variation is computed yearly, with one number per year. However, these figures can be compared with the figures for 2018–2019 presented in the S1 Appendix.

Regarding age, the following observations were made:

No changes in healthcare-seeking behavior were noted across age groups before the political measures against the COVID-19 virus (P1).

During the lockdown ($P_2$), we observe a global decrease in healthcare behavior regardless of age.

Heterogeneity in healthcare behavior increased during subsequent periods (P3 and P4). For period $P_3$, signs diverged based on age group, however, the absolute value of figures remained comparable to a situation without a COVID-19 pandemic (compared with the years 2018–2019, see S1 Appendix).

During the shutdown ($P_4$), the oldest patients were less frequently admitted in 2020 compared to 2019. Notably, we observe a stronger decrease for the group over 85 years old. Groups below 75, namely working people and young seniors, returned to the same healthcare-seeking behavior as before the COVID-19 pandemic. There was possibly even a little catch-up in admission rates during the "in-between" period. Whereas, above 75 and especially above 85 years old, a continued decrease in healthcare-seeking behavior for stroke continued to be observed. It is important to note that an ischemic stroke requires that one be driven to the nearest emergency room.

For stroke patients with chronic diseases, the variation is essential both during the lockdown period ($P_2$) with a -13% decrease and to some extent during the shutdown period ($P_4$) with a -5% decrease. Patients without complications were much more likely to renounce (-11%) being admitted for stroke during the lockdown period ($P_2$) than those with complications. Also, the group of patients without consequences exhibited a -14% variation in not being admitted for stroke during the lockdown period ($P_2$). These results suggest a potential loss of opportunity for these patients. In addition, we observed a decrease in patients admitted

with consequences (-8%) during the lockdown period ($P_2$). These results raise the critical question of loss of opportunity.

Concerning geographic localization of the patient's residence, t similar patterns were observed regardless of locale type (rural areas, isolated cities, downtown areas, suburban areas). What was observed was a strong lockdown effect ($P_2$) with a variation in absolute value above 10% and a negative sign; an "in-between period" ($P_3$) where patients returned to their 2019-healthcare-seeking behavior except in downtown areas, where the rebound is slower; The shutdown period ($P_4$) had a weak or even no effect on healthcare seeking behavior except in the suburban areas, with a decrease rate of -5%.

Turning to individual vulnerability conditions, we analyze the most extreme vulnerability index. According to this index, there was little to no variation in the number of highly vulnerable inpatients from 2019 to 2020. Contrary to what could be expected, we do not observe any decrease in behavior coding or a lower number of these inpatients. We do observe that the number of highly vulnerable patients for stroke is not negatively impacted by the lockdown period ($P_2$), or more precisely, we observe an increase in the number of these inpatients admitted in 2020 as compared to 2019. For individuals not if extremely vulnerable, we find what was globally observed for all inpatient admissions (Fig 1): a substantial decrease in admission during the lockdown period ($P_2$), a return to the healthcare behavior as in 2019 during the "in-between" period ($P_3$), and a weak effect of the shutdown period ($P_4$) on healthcare seeking behavior.

Examining the French deprivation index (Fdep), no changes in healthcare-seeking behavior were detected during the first period ($P_1$), compared to 2019. (We probably overinterpret when saying that the more well-off population anticipated the COVID-19 pandemic consequences by reducing their healthcare admission behavior). however, during the lockdown period ($P_2$), patients renounced seeking healthcare compared to 2019. The gradient of deprivation followed the U-shape in the variation of admission, where both the better-off and the most-deprived groups showed a much more marked decrease. The "in-between" period ($P_3$) was quite similar to the analogous 2019 period but the better-off group started to catch up, while the worst-off began to lag behind. The shutdown period ($P_4$) was characterized by heterogeneity of healthcare-seeking behavior: the deprived ones (from $Q_3$) stood firm on being less admitted than before the COVID-19 pandemic.

### 4.3 The structural analysis

As previously presented, the dependent variable $Y_{ij}$ is the variation in healthcare access. The greater the variation is, the more important the healthcare renouncement is. As explained in Section 3.1 "A non-parametric model", the subgroup$_{ij}$ is determined by specific clinical, demographic, vulnerability, and deprivation factors, which establish the group $i$ for a given period $j$.

We present five models (Model A to Model E) to study the potential correlated effects. The main results are presented in Table 2. Detailed information is available (Table A2 to A5) in S1 Appendix.

During the pre-COVID-19 pandemic period ($P_1$), results exhibit no significant change in healthcare access behavior. Groups of individuals suffering from stroke exhibited similar patterns of behavior in 2019 and 2020 during this period $P_1$. This result is found when comparing groups with stroke complications to those without complications, as well as groups with stroke consequences to those without consequences. Additionally, no difference was observed between people with stroke living in a rural area versus an urban one in terms of healthcare access behavior. Turning to vulnerability and deprivation, the level of deprivation does not affect the result (Model A, Model B, Model C). However, we observe an increase in admission

**Table 2. Regression coefficients on variation in inpatients' healthcare-seeking behavior per 10'000 inhabitants.**

| | Pre-COVID-19 pandemic period | | | During the lockdown period | | | |
|---|---|---|---|---|---|---|---|
| | Model A | Model B | Model C | Model A | Model B | Model C | Model E |
| Vulnerability index | -5.618* (0.086) | -7.000* (0.317) | -5.280* (0.065) | -26.330*** (0.000) | 25.125*** (0.000) | -25.887*** (0.000) | -22.872*** (0.000) |
| Deprivation: Better-off (Q1) | 3.191 (0.194) | | | -2.033 (0.202) | | | |
| Deprivation: the deprived (Q4 & Q5) | | -1.982 (0.719) | | | -1.364 (0.143) | | |
| Deprivation: the most deprived (Q5) | | | -3.508 (0.179) | | | +4.442 (0.210) | +2.032 (0.450) |
| Regional crisis | | | | | | | 7.125*** (0.000) |
| Constant | 12.437*** (0.000) | 16.953*** (0.001) | 17.365*** (0.000) | 93.232*** (0.000) | 82.241*** (0.000) | 88.862*** (0.000) | 74.997*** (0.000) |
| R-squared (%) | 43.53 | 30.31 | 32.13 | 14.97 | 18.16 | 21.31 | 26.55 |
| | "In between" period | | | During the shutdown period | | | |
| | Model A | Model B | Model C | Model A | Model B | Model C | Model E |
| Vulnerability index | -2.678 (0.351) | -1.859 (0.452) | -2.493 (0.321) | -6.674* (0.073) | -5.982* (0.083) | -6.455 (0.101) | -2.128* (0.059) |
| Deprivation: Better-off (Q1) | -2.368 (0.409) | | | -9.655* (0.081) | | | |
| Deprivation: the deprived (Q4 & Q5) | | 3.508* (0.065) | | | -1.982 (0.700) | | |
| Deprivation: the most deprived (Q5) | | | 4.522* (0.071) | | | 6.746* (0.061) | 3.046*** (0.002) |
| Regional crisis | | | | | | | 4.935*** (0.000) |
| Constant | 9.023 (0.019) | 10.861 (0.001) | 9.473 (0.006) | 29.379 (0.000) | 26.246 (0.000) | 27.701 (0.000) | 10.723 (0.000) |
| R-squared (%) | 15.85 | 25.11 | 24.91 | 14.97 | 18.16 | 21.31 | 26.55 |

Source: PMSI-MCO, INSEE-IRCOM, INSEE-INSERM, exhaustive administrative database for stroke patients _ 2019–2020.

Note: Each row reports regression coefficients from a linear regression model, weighted by individuals at risk of having a stroke a given period, in year 2019. P-values are in parentheses. In addition to the listed variables, we control for age, stroke complications, stroke consequences, and rural area. For Model D, we also control for chronic disease information, and Model E we control for regional crisis dummy. Standard errors are clustered at the region level. As robustness checks, we controlled for region fixed effects. Results are similar.

*: $p < .1$;

**: $p < .05$;

***: $p < .01$.

compared to the others for the most vulnerable individuals compared with 2019. Similarly, groups of stroke patients with chronic diseases were admitted more frequently than others compared to 2019.

During the lockdown period, there was a decline in hospital admissions for stroke. The magnitude of the intercept indicates a significant impact on healthcare access to hospitals for individuals with stroke. Yet, the population was impacted in a similar way by political measures intended to restrain COVID-19's effects on public health. Notably, there was no difference in healthcare access between urban and rural areas for individuals with stroke. For regions more severely impacted by the COVID-19 virus spread (as captured by the regional crisis dummy variable), a more substantial decrease in healthcare access was observed compared to stroke individuals living in the other French regions (Model E).

Next, we show that the deprivation level did not affect the change in healthcare access (Model A, Model B, Model C). In addition, for groups of patients with chronic diseases, the changes in healthcare access from 2019 to 2020 were similar to those without chronic diseases (Model D). However, we revealed a positive effect on healthcare access behavior for the most vulnerable groups: extremely vulnerable subgroups were more likely to be admitted than other groups compared to the reference year, 2019. This suggests that the most vulnerable stroke patients seem to have been protected from being shielded from (or "shut out of") the healthcare system.

In addition, groups of patients suffering complications and stroke consequences are also less likely to have access to healthcare. Effective hospital care management of stroke patients without complications and consequences is crucial to increase their likelihood of a returning to normal. This result underscores the issue of loss of opportunity for these patients without complications or consequences. During the "in-between" period, we observe heterogeneity in the change of healthcare-seeking behavior. Groups of individuals with stroke complications are relatively less likely to forgo healthcare access. Concerning stroke patients, the elderly population (over 74 years old) had less healthcare access than they had in 2019, compared to younger groups. In the same vein, the change in healthcare access for stroke victims living in rural areas remained comparable to those in urban areas, suggesting a lag in the return to pre-COVID-19 pandemic healthcare access for rural residents.

To be in dire need does not appear to influence the change in healthcare access for stroke sufferers, using the year 2019 as a benchmark. On the contrary, we find that when poor people are not desperately vulnerable, these poor people (belonging to the last quintile of deprivation) are actually more likely to renounce healthcare access compared to others, with the year 2019 being the benchmark (Model B and Model C).

The negative sign for better-off subgroups suggests that they were more likely to revert to their pre-pandemic behavior (Model A). However, the effect is not statistically significant.

During the shutdown period, we observed a notable impact of age on the change in healthcare access, with older stroke patients being more likely to experience reduced access compared to the younger stroke population. In other words, the most aged are the most affected in terms of diminished healthcare access.

In terms of clinical conditions, we found that stroke patients with complications experienced fewer changes to obtain healthcare access compared to others. However, we did not find any difference in the chronic disease stroke group compared to others, nor between the groups of stroke patients with consequences and those without consequences.

Geographical residence, either in a rural or urban area, did not appear to be a significant driver. However, living in a region heavily affected by the spread of the virus was a driver for a larger reduction in healthcare access.

To be vulnerable negatively influenced the change in healthcare access for stroke sufferers, using the year 2019 as a benchmark. It suggests once again that the public healthcare system protects the most extremely economically-vulnerable segment of the population. They do not renounce healthcare access in the same proportion. However, this result is not as powerful as during the lockdown period. The magnitude and significance of the coefficients (from Model A to Model E) are weaker.

Furthermore, during the shutdown, better-off groups of stroke patients changed their healthcare access less than those in more economically deprived situations (Model A). The results of Model B and Model C suggest that the deprivation gradient followed the reduction of healthcare access compared to those in better economic situations. Table 3, presents the DiD estimates based on Eq (2). We then assess the potential differential effect of the socioeconomic gradient in areas of regional crisis.

**Table 3. Regression coefficients on variation in inpatients' healthcare-seeking behavior per 10'000 inhabitants.**

| | During the lockdown period | | | During the shutdown period | | |
|---|---|---|---|---|---|---|
| | Model A' | Model B' | Model C' | Model A' | Model B' | Model C' |
| Vulnerability index | -20.236*** (0.000) | -17.256*** (0.006) | -18.934*** (0.000) | -6.816*** (0.002) | -4.022* (0.088) | -5.490*** (0.009) |
| Deprivation: Better-off (Q1) | -1.781*** (0.000) | | | -6.876** (0.013) | | |
| Deprivation: the deprived (Q4 & Q5) | | 2.608 (0.108) | | | -0.0843 (0.975) | |
| Deprivation: the most deprived (Q5) | | | 1.628 (0.270) | | | 4.368*** (0.004) |
| Regional crisis | 8.839*** (0.002) | 9.086* (0.071) | 10.706*** (0.000) | 4.160* (0.058) | 2.420 (0.298) | 2.351 (0.258) |
| *Regional crisis crossed* | | | | | | |
| Vulnerability index | 5.252*** (0.006) | 6.795*** (0.001) | 6.380*** (0.010) | 4.9397 (0.193) | 3.413 (0.901) | 2.752 (0.382) |
| Deprivation: Better-off (Q1) | 2.9277 (0.502) | | | -3.349 (0.375) | | |
| Deprivation: the deprived (Q4 & Q5) | | 1.9264 (0.292) | | | -.3817 (0.909) | |
| Deprivation: the most deprived (Q5) | | | 4.7165** (0.054) | | | 0.7431 (0.814) |
| Constant | 31.334*** (0.000) | 26.309*** (0.000) | 29.760*** (0.000) | 7.7415*** (0.003) | 4.147* (0.105) | 5.158** (0.033) |

Source: PMSI-MCO, INSEE-IRCOM, INSEE-INSERM, exhaustive administrative database for stroke patients _ 2019–2020.

Note: Each row reports regression coefficients from a linear regression model, weighted by individuals at risk of having a stroke a given period, in year 2019. P-values are in parentheses. In addition to the listed variables, we control for age, stroke complications, stroke consequences, rural area, chronic disease information, and regional crisis. Standard errors are clustered at the region level. As robustness checks, we controlled for region fixed effects. Results are similar.

*: $p < .1$;

**: $p < .05$;

***: $p < .01$.

The analysis indicates that the most vulnerable, including those who are often excluded from public statistics due to precarious situations (such as homelessness), experience a negative impact on access to stroke care during the lockdown period. Nevertheless, the coefficient becomes positive in regions with a specific pandemic peak. This result suggests that the healthcare system protected vulnerable individuals overall but to a lesser extent in COVID-19 pandemic regions. There was no significant effect between the regions most affected by the pandemic and the other regions during the shutdown. However, in these hardest-hit regions, the effect on the healthcare renouncement rate is positive.

We then investigated whether the effects of the deprivation index differed across regions. Initially focusing on the better-off stroke patients, we find that the lack of difference in behavior regardless of the level of deprivation index (Table 2, model A) is due to regional differences in the COVID-19 pandemic. The results reveal that the better-off were less likely to forego care compared to other socioeconomic groups. However, this healthcare-seeking behavior disappeared in the COVID-19 pandemic regions (Table 3, Model A').

For the most disadvantaged (Table 3, Model C'), the results showed that in the COVID-19 pandemic regions, renunciation of care was even more marked during the lockdown period.

## 5. Discussion

In this paper, we have examined the distributional consequences of the COVID-19 pandemic on stroke patients, using a study sample of 214,720 patients treated for ischemic strokes. This paper lends support to calls for distinguishing periods over the year 2020. The lockdown period had not affected healthcare-seeking behavior as the shutdown period did. If healthcare-seeking behaviors were quite homogeneous over the first lockdown period, for the periods that followed, the behavior pattern was much more heterogeneous.

Indeed, the initial lockdown period, healthcare-seeking behavior was impacted similarly for everyone. However, when we examine regional differences, the results reveal that the most affluent began to forgo care only in the most affected regions. In these same COVID-19 pandemic regions, the most disadvantaged forewent care even more. Furthermore, stroke patients with complications and consequences renounced going to the hospital less than others. The inpatients in whom thrombolytics could have been administrated are probably among those who most renounced access during the first lockdown. This result suggests an actual loss of opportunity.

The "in-between" and the shutdown periods were characterized by heterogeneity in healthcare-seeking behavior. Patients over 75 years old were notably affected, suggesting age as a driver to explain the reduction in the number of individuals seeking health care. Additionally, patients in the top quintile of the wealth distribution were less impacted by the healthcare access renouncement than those in the bottom quintile. It suggests that after an exogenous shock affecting the whole population, the upturn in behavior to levels seen previous to the COVID-19 pandemic period depends on deprivation factors and age threshold. These results are in line with those of [38–41]. During the early containment phase of COVID-19 at Hong Kong, Teo et al. show a prolongation in stroke onset to hospital arrival time and a significant reduction in individuals arriving at the hospital within 4.5 hours and presenting with transient ischemic attack. Leira et al. [42] explained that the coronavirus 2019 (COVID-19) pandemic required drastic changes in allocation of resources, which affected the delivery of stroke care. As well, Montaner et al. (2020) showed that the COVID-19 pandemic was disruptive for acute stroke pathways.

Here we explore the effect of the pandemic over the year 2020 on healthcare demand behavior. Using an administrative database including all inpatients for stroke over the years 2019 and 2020, we have obtained an exhaustive picture of the impact the 2020 pandemic had on these patients. The reasonable hypothesis that we formulate is that stroke patients are not concerned by a hospital unit's shutdown for COVID-19 reasons because of the stroke patient's urgent need for care. Therefore, the reduction in healthcare access is a proxy for healthcare renouncement. By extension, a study on purely elective care will complete the analysis of the impact of the COVID-19 pandemic's political measures on non-COVID-19 individuals for healthcare-seeking behavior. Such a study implies using a longer period, such as 2019 to 2021.

The effect of COVID-19 and restrictions on the probability of stroke is mixed with the impact on care utilization. In this paper, we identify individuals admitted to the hospital for stroke. We observed hospitalizations for stroke throughout 2019 and 2020 (2018 was used for sensitivity analyses). It is conceivable that the marked decrease early in the strict containment period is due to an external effect of policy-driven COVID-19 restrictions on the probability of having a stroke. However, we observed a decrease before the strict confinement period, i.e., before a lifestyle change. At the same time, we observe a slow rebound in stroke admissions before the end of confinement, i.e., when the population was still in a unique situation, constrained in their daily movements. Moreover, in France in 2020, the policy decisions that restricted individual's travel were identical regardless of location. Furthermore, we show a differential effect according to the deprivation index after the lockdown period.

Besides, the study based the calculation of four periods on government measures instead of the progression of the pandemic. Although the French population had access to daily figures presented by the government, there is a possibility that some groups did not modify their behavior based on the advancement of the pandemic, but rather on the changing social measures.

Further analyses may be necessary to understand the information's imperfections and other biases that may limit or impede healthcare-seeking behavior for some groups, particularly when a shutdown is implemented during the COVID-19 pandemic.

In addition, we opted to present the outcomes with the "regional crisis" factor to ensure consistency between the political decisions identifying peak crisis periods and those that define regions in a health crisis, from the regulator's perspective. However, in our preliminary work, we used COVID hospitalization rate per 1,000 inhabitants at the department level. The main results presented in this paper were also derived using this approach. Additionally, for this study, we used 2018 and 2019 data as a benchmark. To enhance model robustness, it would have been interesting to extend the time frame and evaluate the sensitivity of the outcomes.

## 6. Conclusion

We investigated whether political decisions to contain the spread of the COVID-19 virus resulted in missed opportunities and inequalities in healthcare access. The results we present here might decisively trigger an alert on it.

What happened to the missing inpatient stroke events hospitalized compared to 2019 is the puzzling question raised as a conclusion of this paper. There is no way to track those who did not seek care. This remains an open question. We may track hospital admissions correlated with former stroke event(s) in the future.

Thus, our results have serious implications for public health policy. They underscore the evidence-based socioeconomic and age-related disparities in healthcare access in a system that operates under the mandate of providing equal access based on need. Our study suggests a loss of opportunities during the lockdown and an increase in inequality during the periods that followed this lockdown. It also highlights disparities in the impact of political decisions on the non-COVID-19 population requiring healthcare, from lockdown to shutdown phases. Equity concerns are particularly pressing in a publicly-funded healthcare system., where equal access to care based on need is the core of such a system.

## Supporting information

**S1 Appendix.**
(DOCX)

## Author Contributions

**Data curation:** Marie Moitry.

**Formal analysis:** Carine Milcent, Hanta Ramaroson, Florence Binder-Foucard.

**Methodology:** Carine Milcent, Hanta Ramaroson, Marie Moitry, Anne-Marie Moulin.

**Software:** Fleur Maury.

**Supervision:** Carine Milcent.

**Validation:** Carine Milcent, Fleur Maury, Florence Binder-Foucard, Marie Moitry, Anne-Marie Moulin.

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
