## [Decision Letter · Decision Letter 0]

6 Feb 2024

PONE-D-23-34429Stroke but No Hospital Admission Lost Opportunity for Whom?PLOS ONE

Dear Dr. Milcent,

Thank you for submitting your manuscript to PLOS ONE. After careful consideration, we feel that it has merit but does not fully meet PLOS ONE’s publication criteria as it currently stands. Therefore, we invite you to submit a revised version of the manuscript that addresses the points raised during the review process.

We look forward to receiving your revised manuscript.

Kind regards,

Bruno Ventelou

Academic Editor

PLOS ONE

Journal Requirements:

Additional Editor Comments:

Sorry for the long delay; due to technical problems with a referee report (that was hard to access for).

Reviewers' comments:

Reviewer's Responses to Questions

**Comments to the Author**

1. Is the manuscript technically sound, and do the data support the conclusions?

Reviewer #1: Yes

Reviewer #2: Yes

2. Has the statistical analysis been performed appropriately and rigorously? 

Reviewer #1: Yes

Reviewer #2: No

3. Have the authors made all data underlying the findings in their manuscript fully available?

Reviewer #1: No

Reviewer #2: Yes

4. Is the manuscript presented in an intelligible fashion and written in standard English?

Reviewer #1: Yes

Reviewer #2: Yes

5. Review Comments to the Author

Reviewer #1: Dear authors,

Thanks for this article. Here are my comments, highlighting specific points in your manuscript.

***

Computed variables: “From the patient’s postcode, we set up a dummy for residences in a rural area.”

Could you explain why you chose to include this very specific variable even though you also include a variable about the deprivation index?

***

Computed variables: “A dummy for enrollment in universal health protection, called PUMa, is defined. This insurance (Protection Universelle Maladie) is part of public health insurance. It is designed for people outside the labor market and no longer seeking employment.”

Could you also explain why you chose to include this variable even though you included a variable about the deprivation index and vulnerability?

***

Computed variables: “The extreme vulnerability dummy variable is defined as having at least one of these individual characteristics”. I do not understand why you decide not disentangling between people with one characteristic and people two or more characteristic.

Could you at least present the distribution among the people you consider in your analysis?

***

Specific computed variables for the Covid-19 pandemic: “In France, the COVID-19 pandemic can be divided into four periods over 2020 (P1 to P4), according to intensity.”

Why did you choose to use the four periods calculated based on government measures rather than considering the evolution of the pandemic? People in France were aware of all the figures, presented each day by the government, and, likely, they did not adjust their behavior based on the progression of waves as much as the evolution of social measures. Could you discuss this point?

***

Specific computed variables for the COVID-19 pandemic: “We created a binary variable by period (regional crisis) equal to one when the patient’s residence was in a critical situation area.”

Could you precisely explain the indicator you have used to classify the French regions? Additionally, this classification is based on epidemic indicators, which is different from your definitions of the sub-periods.

***

Empirical framework: “This paper assumes that, without the COVID-19 pandemic, stroke patient distribution in 2020 should have followed stroke patient distribution in 2019. As a comparative analysis, we checked that this is observed when comparing 2018 to 2019. The method described below has been used in the years 2018-2019. The differences in magnitudes of in-admission rates from 2018 to 2019 are negligible. These differences are used as a benchmark to judge the 2019-2020 differences. “

Could you enlarge your time window to assess the validity of this argument; maybe use data before 2018? 2019 was a very specific year for the evolution of life expectancy in France, and even if it is not visible for strokes, I would appreciate it if you could discuss this point.

***

Empirical framework: “The construct is the following: we set up groups of patients i according to a set of variables.“

Could you be more precise about the construction of all the groups? You should introduce in this paragraph the explanations you give in the results sections.

***

Results: “As sensitivity analysis, we run models with alternative sets of variables to set up groups of patients. Results are available upon request.“

Could you be more precise about the construction of these alternative groups, and maybe present the results in the appendix?

***

Discussion: “It suggests that after an exogenous shock affecting the whole population, the upturn in behaviour to levels seen previous to the COVID-19 pandemic period depends on deprivation factors and age threshold. I am quite sure that there exists a large literature about this question.“

Could you cite this literature in the discussion? Less specifically, I am a little be surprised that no article is quoted in the discussion part.

***

Tables: In Table 1, I do not understand the definition of the variable in the last column. Moreover, I am sure that you could present these results with a figure and not in a table.This should help the readers to easily grasp the magnitude of your results.

Reviewer #2: > As regards the content and the statistical analysis (point 2. above):

It is not clear if a panel methodology was used. Colinearity of the explanatory variables was not checked. Post-tests on residuals were not run. If panel estimation was carried out, the Hausman test was not run.

> As regards the form of the article (Point 4. above), some sentences may be clarified for better understanding, such as:

- Page 4: In « In this paper, using a very innovative method », it woud be useful to precise of which innovative method the authors are referring to, from the start of the paper. In section « 3.1 A non-parametric method » more details could be added to present the model and the methods used, if the character count makes it possible.

- Page 7: In section « 3.1 A non-parametric method », in « We used a non-parametric method », the authors coul precise to whic method they are referring to.

- Page 7: In « The subgroupij is determined by specific clinical, demographic, vulnerability, and deprivation factors setting up the group i for a given period j », is it possible to mention, how the subgroups are constituted, with which thresholds for example ?

- Page 8: In « We introduce weightings in the model », the authors may briefly detail which kind of weightings is used or how they are built.

6. PLOS authors have the option to publish the peer review history of their article (what does this mean?). If published, this will include your full peer review and any attached files.

Reviewer #1: No

Reviewer #2: No

---

## [Decision Letter · Decision Letter 1]

2 Jul 2024

Stroke but No Hospital Admission Lost Opportunity for Whom?

PONE-D-23-34429R1

Dear Dr. Milcent,

We’re pleased to inform you that your manuscript has been judged scientifically suitable for publication and will be formally accepted for publication once it meets all outstanding technical requirements.

Kind regards,

Bruno Ventelou

Academic Editor

PLOS ONE

Additional Editor Comments (optional):

thanks for your work

Reviewers' comments:

Reviewer's Responses to Questions

**Comments to the Author**

1. If the authors have adequately addressed your comments raised in a previous round of review and you feel that this manuscript is now acceptable for publication, you may indicate that here to bypass the “Comments to the Author” section, enter your conflict of interest statement in the “Confidential to Editor” section, and submit your "Accept" recommendation.

Reviewer #1: All comments have been addressed

Reviewer #2: All comments have been addressed

2. Is the manuscript technically sound, and do the data support the conclusions?

Reviewer #1: Yes

Reviewer #2: Yes

3. Has the statistical analysis been performed appropriately and rigorously? 

Reviewer #1: Yes

Reviewer #2: Yes

4. Have the authors made all data underlying the findings in their manuscript fully available?

Reviewer #1: Yes

Reviewer #2: (No Response)

5. Is the manuscript presented in an intelligible fashion and written in standard English?

Reviewer #1: Yes

Reviewer #2: Yes

6. Review Comments to the Author

Reviewer #1: Dear authors,

Thank you very much; I am satisfied with the answers you have given to my queries.

Kind regards.

Reviewer #2: (No Response)

7. PLOS authors have the option to publish the peer review history of their article (what does this mean?). If published, this will include your full peer review and any attached files.

Reviewer #1: **Yes: **Dr. Florian Bonnet

Reviewer #2: No

---

## [Editor Report · Acceptance letter]

11 Jul 2024

PONE-D-23-34429R1 

PLOS ONE

Dear Dr. Milcent, 

I'm pleased to inform you that your manuscript has been deemed suitable for publication in PLOS ONE. Congratulations! Your manuscript is now being handed over to our production team.

Kind regards, 

on behalf of

Dr. Bruno Ventelou 

Academic Editor

PLOS ONE